# Modified Histopathological Protocol for Poly-ɛ-Caprolactone Scaffolds Preserving Their Trabecular, Honeycomb-like Structure

**DOI:** 10.3390/ma15051732

**Published:** 2022-02-25

**Authors:** Tomasz Dębski, Juliusz Wysocki, Katarzyna Siennicka, Jakub Jaroszewicz, Karol Szlązak, Wojciech Święszkowski, Zygmunt Pojda

**Affiliations:** 1Department of Regenerative Medicine, Maria Sklodowska-Curie National Research Institute of Oncology, Wilhelma Konrada Roentgena 5, 02-781 Warsaw, Poland; jwysocki@coi.pl (J.W.); siennicka.katarzyna@wp.pl (K.S.); zygmunt.pojda@coi.pl (Z.P.); 2Materials Design Division, Faculty of Materials Science and Engineering, Warsaw University of Technology, Woloska 141, 02-507 Warsaw, Poland; jakubjaroszewicz@wp.pl (J.J.); karolszlazak@wp.pl (K.S.); wojciech.swieszkowski@pw.edu.pl (W.Ś.)

**Keywords:** PCL, scaffold, histopathological protocol, scaffold architecture preservation, full cross-section of scaffold

## Abstract

Poly-ɛ-caprolactone (PCL) is now widely studied in relation to the engineering of bone, cartilage, tendons, and other tissues. Standard histological protocols can destroy the carefully created trabecular and honeycomb-like architecture of PCL scaffolds, and could lead to scaffold fibers swelling, resulting in the displacement or compression of tissues inside the scaffold. The aim of this study was to modify a standard histopathological protocol for PCL scaffold preparation and evaluate it on porous cylindrical PCL scaffolds in a rat model. In 16 inbred Wag rats, 2 PCL scaffolds were implanted subcutaneously to both inguinal areas. Two months after implantation, harvested scaffolds were first subjected to μCT imaging, and then to histopathological analysis with standard (left inguinal area) and modified histopathological protocols (right inguinal area). To standardize the results, soft tissue percentages (STPs) were calculated on scaffold cross-sections obtained from both histopathological protocols and compared with corresponding µCT cross-sections. The modified protocol enabled the assessment of almost 10× more soft tissues on the scaffold cross-section than the standard procedure. Moreover, STP was only 1.5% lower than in the corresponding µCT cross-sections assessed before the histopathological procedure. The presented modification of the histopathological protocol is cheap, reproducible, and allows for a comprehensive evaluation of PCL scaffolds while maintaining their trabecular, honeycomb-like structure on cross-sections.

## 1. Introduction

Poly-ɛ-caprolactone (PCL) is a hydrophobic, semi-crystalline polymer which was synthesized in the 1930s [1]. It undergoes a two-step degradation process. The first stage is the non-enzymatic hydrolytic cleavage of ester groups, whereas the second is dependent upon polymer crystallinity and is described as intracellular degradation [2,3]. The in vitro degradation of PCL is followed by an increase in the polymer’s degree of crystallinity [4,5]. The increase in crystallinity is governed by crystalline fibril reorganization, which affects PCL’s degradation rate [3]. PCL is nontoxic and has a suitable surface for cell proliferation and differentiation, and its non-toxic degradation products are usually metabolized and eliminated via natural pathways [6].

Exceptional properties, such as biocompatibility, the type of biodegradation, and good mechanical and physicochemical parameters [7,8] have stimulated extensive research into the potential applications of PCL in the biomedical field. For decades, PCL was widely used as a component of medical sutures [9], wound dressings [10,11], contraceptive devices [12], fixation devices [13], and filling materials in dentistry [14,15]. Recently, thanks to the development of tissue engineering, attention has again been drawn to PCL. The rheological and viscoelastic properties [16], tailorable degradation kinetics, mechanical properties, ease of shaping and fabrication, enabling appropriate cell-friendly pore size creation, inexpensive production process, and FDA approval renders PCL easy to manufacture and manipulate into a broad range of scaffolds. Moreover, the development of various fabrication technologies such as solid free-form fabrication, fused deposition modeling, electrospinning nanofibers, etc., [17,18,19] have allowed the use of PCL for producing 3D scaffolds of various shapes and with customized inner architecture suiting a specific anatomical site. PCL is now widely studied in the engineering of bone [20], cartilage [21], tendons and ligaments [22], heart valves [23], blood vessels [24], nerves [25], and other tissues. 

Extensive use of PCL scaffolds in tissue engineering and the rapid development of new fabrication methods allows the mimicking of 3D architecture of various engineered tissues, revealing the need for the modification of standard histological protocols to derive a more detailed view for the morphological analysis of explanted scaffolds.

Standard histological protocols [26] can destroy the carefully created 3D, honeycomb-like architecture of PCL scaffolds and lead to scaffold fiber swelling, resulting in the displacement or compression of tissues inside the scaffolds. 

Modification of these protocols is needed to facilitate the evaluation of the whole scaffold cross-section and the spatial relationships inside the scaffold, visualize scaffold pores with tissue ingrowth, analyze the interaction between the scaffold and the surrounding tissue, or assess the scaffold degradation rate. 

The aim of this study was to modify a standard histopathological protocol for PCL scaffold preparation and evaluate it on porous cylindrical PCL scaffolds in a rat model. 

The biocompatibility of the tested scaffolds was proven in our previous study on an animal model, where we found that adipose-derived stem cells (ASCs) seeded into PCL scaffolds promoted proliferation, adhesion, and osteogenic differentiation [27]. 

To the best of our knowledge, this is the first report in the literature presenting a histopathological protocol which preserves the trabecular, honeycomb-like architecture of PCL scaffolds and more detailed and reliable cross-section views. 

## 2. Materials and Methods

### 2.1. Scaffold Fabrication

A porous cylindrical scaffold (10 × 6 mm) with an empty internal space (6 × 2 mm), was designed with SolidWorks 2012 (Waltham, MA, USA) CAD software (Figure 1a). The fiber pattern was repeated every 5 layers until 50 layers were contained in each scaffold (Figure 1b).

Next, 32 poly-ε-caprolactone (PCL) scaffolds were fabricated layer-by-layer with a 3D printer (Bioscaffolder, SYS+ENG, Salzgitter-Bad, Germany) using the fused deposition modelling technique, with the settings shown in Table 1 [28]. 

Before the experiments, the shape fidelity, fiber layer architecture, morphology, and porosity of the 3D-printed PCL scaffolds were observed using μCT. 

Prior to implantation, the scaffolds were sterilized in 70% ethanol.

### 2.2. In Vitro Initial Studies

During the initial studies, we identified two factors—xylene and temperature—that may affect the structure of the scaffold during the standard histopathological procedure. 

To evaluate the scaffold, microscopical images were taken using an Olympus CKX41 microscope after xylene addition at a temperature of 23 °C and after bathing in water at a temperature of 57 °C for 10 min, accordingly. Scaffold structure, fiber diameters, and pore sizes were observed. 

### 2.3. Animal Study Design

In the animal model (16 inbred WAG rats, aged 4–6 months, weighing 240–280 g), two PCL scaffolds were implanted subcutaneously to both inguinal areas of each rat. All animal experiments were approved by the II Local Bioethical Committee in Warsaw, Poland (Approval number: 31/2011), and performed according to the Guidelines for the Regulation of Animal Experiments.

Animals were anesthetized by the intramuscular injection of medetomidine (0.2 mg/kg), ketamine (20 mg/kg), and butorphanol (1 mg/kg). After anesthesia, the inguinal areas of the rats were epilated and sterilized. Next, through a 2 cm incision, the PCL scaffold was implanted to the pocket created subcutaneously and the wound was closed using interrupted absorbable sutures (Vicryl 3-0) (Figure 2).

After surgery, the animals were administered postoperatively with analgesics and remained under constant observation until they were awakened. After 2 months (standard implantation time for tissue-engineered products), the rats were sacrificed by the intraperitoneal administration of phenobarbital (200 mg/kg body weight) and the scaffolds were harvested. 

The harvested scaffolds were fixed in 10% buffered formalin and subjected first to μCT imaging and then to histopathological analysis. A standard and modified histopathological protocol was performed on the scaffolds harvested from the right and left inguinal areas, respectively. 

### 2.4. Histopathological Analysis

#### 2.4.1. Standard Histopathological Protocol

The samples were dehydrated using graded ethanol. Clearing was performed using xylene (Xylene (isomeric mixture), Sigma-Aldrich, Munich, Germany). All samples were embedded in paraffin blocks at 57 °C using standard paraffin with a melting point of 56–58 °C. (Paraffin, Chempur, Piekary Slaskie, Poland). Next, the mid-part of each scaffold (half of the implant height) was serially sectioned into sections 4 μm in thickness using a rotary microtome (Leica RM2245, Leica, Wetzlar, Germany). Obtained sections were transferred onto standard microscope slides (Microscope slide, Zarys, Zabrze, Poland). The slides were deparaffinized with xylene and rehydrated.

The slides were stained with hematoxylin and eosin (HE) according to standard procedures (Sigma-Aldrich, Munich, Germany). After subsequent dehydration, the sections were cover-slipped using xylene-based glue (Consul-Mount, Thermo Scientific, Waltham, MA, USA) (Table 2).

Slides from the middle of each scaffold were subjected to further analyses.

#### 2.4.2. Novel Histopathological Protocol

The samples were dehydrated using graded ethanol. In the clearing stage, xylene was replaced by a d-limonene (HistoClear^®^, National Diagnostics, Atlanta, GA, USA), to avoid distortion of the scaffold fibers (see Discussion). Low melting point paraffin (Paraffin 46–48 in block form, Sigma-Aldrich, Munich, Germany) was used for embedding all samples in paraffin blocks at 49 °C. The blocks containing the mid-part of each scaffold (half of the implant height) were serially sectioned into sections 4 μm in thickness using a rotary microtome (Leica RM2245, Leica, Wetzlar, Germany). Subsequently, the sections were transferred onto highly adhesive microscope slides (Superfrost Ultra Plus^®^, Menzel Glaser, Braunschweig, Germany). The slides were deparaffinized with d-limonene and rehydrated.

The slides were stained with hematoxylin and eosin (HE) according to standard procedures (Sigma-Aldrich, Munich, Germany). 

After subsequent dehydration, the sections were cover-slipped using Canadian balm (Sigma-Aldrich, Munich, Germany) dissolved in d-limonene as the mounting medium (Table 2).

Slides from the middle of each scaffold were subjected to further analyses. 

### 2.5. Scaffold μCT Imaging

μCT imaging was performed for the scaffolds just after fabrication and before implantation to assess the scaffold morphology. After μCT imaging with the settings shown in Table 3, the scaffolds were 3D-reconstructed using NRecon software (Micro Photonics, Inc., Allentown, PA, USA) using post-alignment, ring artifact, and beam-hardening correction. Scaffold porosity, pore size, fiber diameter, fiber layer, and scaffold morphology fidelity were assessed.

Additionally, explanted scaffolds before and after histopathological analysis were subjected to μCT imaging with the settings shown in Table 3.

The μCT images taken in the middle of each scaffold were reconstructed using NRecon software (Micro Photonics, Inc., Allentown, PA, USA) and subjected to further cross-sectional analysis together with the corresponding histopathological slides.

### 2.6. Scaffold Cross-Sectional Analysis

The fixed samples prepared with both histopathological protocols were examined using a Nikon Eclipse TI inverted microscope and, together with the corresponding μCT images, analyzed using Image J software. The obtained cross-sections were analyzed to calculate the scaffold cross-section surface (S1) and soft tissue area inside the scaffold (S2). To standardize the results, the soft tissue percentage (STP) parameter was introduced and defined Equation (1).
STP (%) = S2/S1(1)
where S1 is the scaffold cross-section surface, and S2 is the soft tissue surface.

Equation (1). Soft tissue percentage (STP) as a ratio of the soft tissue surface (S2) to scaffold cross-sectional surface.

### 2.7. Statistical Analysis

Data were collected in Excel (Microsoft, Redmond, WA, USA). Kruskal–Wallis tests were used to determine the distribution of data, and post hoc Bonferroni tests were used for the analyses. The data were evaluated using Statistica 13 Software (StatSoft, Inc., Tulsa, OK, USA) and GraphPad Prism 8 (San Diego, CA, USA). The results were considered significant at a probability level of *p* < 0.05.

## 3. Results

### 3.1. Scaffold Characterization

All fabricated scaffolds had a relatively uniform shape, morphology, and fiber layer architecture, as revealed by μCT imaging (Figure 3). The mean porosity was 52.4 ± 2%, the average pore size was 240.35 ± 10.4 µm, and the fiber diameter was 238.18 ± 10.4 µm.

### 3.2. In Vitro Initial Studies

After coming into contact with xylene, the scaffold fibers started melting and swelling, filling the pores of the scaffold with melted PCL (Figure 4).

Temperatures close to the glass transition point of PCL (58 °C) resulted in scaffold fiber swelling, expansion, and a decrease in the pore size (Figure 5).

### 3.3. Animal Study

All animals included in the study were followed up for 2 months. Scaffolds were well tolerated by the animals and no wound-related problems or autotomies were observed. At the time of sample harvesting, no excessive scarring, fibrosis, or adhesions were observed (Figure 6).

### 3.4. Histopathological Analysis

All scaffolds were completely covered by fibroconnective tissue. No adverse tissue reactions or marked inflammatory signs were noted. The standard histopathological protocol destroyed the trabecular, honeycomb-like architecture of PCL scaffolds. Scaffold fibers were swollen, and the displacement or compression of tissues inside the scaffold was observed (Figure 7a). HE-stained sections prepared according to the modified histopathological protocol revealed that the scaffold was surrounded by fatty and fibrous tissue. The scaffold trabecular architecture was preserved during the histopathological procedure, and its fibers were surrounded by giant multinucleated histiocytes. Evaluations of the whole scaffold cross-section, the spatial relationships inside the scaffold, the visualization of scaffold pores with tissue ingrowth, analyses of the interaction between the scaffold and the surrounding tissue, or assessments of the scaffold degradation rate were possible (Figure 7b).

### 3.5. Scaffold μCT Imaging

μCT imaging of the middle sections of the explanted scaffolds confirmed the preservation of the scaffold architecture after the modified protocol, and the complete distortion of the spatial architecture after the standard histopathological protocol (Figure 8). No scaffold fibers, porous tissue ingrowth, or interaction between the scaffold and tissue were noted in the group of scaffolds after the standard histopathological protocol. Tissue inside the scaffold was compressed and displaced by the scaffold.

### 3.6. Scaffold Cross-Sectional Analysis

The scaffold cross-section surface (S1) and soft tissue area inside the scaffold (S2) were marked on the histopathological sections (Figure 9) and μCT images (Figure 10 and Figure 11). Analysis of histopathological sections revealed a tenfold higher STP, introduced as an indicator of scaffold architecture preservation, after the modified protocol compared with after the standard protocol (44.03 ± 1.21 vs. 4.72 ± 0.73). Additionally, the μCT analysis confirmed better scaffold architecture preservation after the modified procedure compared with the standard histopathological procedure. The STP calculated on the μCT images after the modified procedure was only 2.5% lower compared with the images obtained before the procedure (43.34 ± 1.59 vs. 45.76 ± 1.29). In contrast, after the standard procedure, the STP was ninefold lower than before the procedure (5.23 ± 2.30 vs. 45.23 ± 1.19).

Comparing the differences in the histopathological sections obtained after the modified procedure with μCT images taken before the procedure, the STP was only 1.5% lower (44.03 ± 1.21 vs. 45.76 ± 1.29). After the standard procedure, that difference was 41.5% lower (4.72 ± 0.73 vs. 45.23 ± 1.19) (Figure 12).

## 4. Discussion

The rapid growing interest in PCL as an excellent material for scaffold fabrication has necessitated the improvement in diagnostic techniques to fully evaluate the increasingly spatial and sophisticated 3D constructs imitating engineered tissues. 

Despite the considerable developments in imaging techniques allowing for comprehensive in vivo and postmortem scaffold evaluation, and even the creation of 3D virtual models [29,30], little attention has been paid to the refinement and updating of histopathological techniques, which remain the basic tool for assessments of scaffold morphology, biocompatibility, and interactions with the surrounding cells. The standard techniques used today only allow for evaluations of selected representative fragments, where the scaffold is often cleared, melted, or swollen, and it is not possible to obtain the full cross-section view with the preserved scaffold’s honeycomb-like structure [31,32]. 

The modified histopathological protocol presented in our study does not have these limitations and preserves the sophisticated trabecular structure of the scaffold on cross-sectional views. Compared with the standard protocol, it also enabled the assessment of almost 10x more soft tissues on the scaffold cross-section. Moreover, the percentage of soft tissues (STPs) was only 1.5% lower than in the corresponding µCT cross-sections analyzed before the histopathological procedure, which may have resulted from the process of tissue shrinking during formalin fixation [33,34].

The obtained results show that this method is effective, reproducible, and easily comparable with other methods. Oberdiek et al., in their study evaluating the osteoconductive properties of PCL and PCL + BCP (biphasic calcium phosphate) scaffolds implanted in rat calvarial defects, used the technique of plastic embedding in Technovit 9100 (Technovit 9100, Kulzer GmbH, Hanau, Germany). Their scaffolds, compared with those used in our study, had a smaller diameter (4 mm vs. 6 mm), a slightly smaller porosity (240 µm vs. 300 µm), and their implantation time included three periods (90, 30, 10 days vs. 60 days). The histopathological images presented in their study did not show the entire cross-section of the scaffold, only its small representative fragments and only the longitudinal cross-section of the calvarial defect filled with the scaffold. The elements of the scaffold visible in these images, in contrast to those obtained in our study, were fused, melted, and swollen. Moreover, the trabecular structure of the scaffold with its fibers arranged in a honeycomb-like structure was not visible [35]. The same histopathological technique was used by Yun et al. They investigated a bone-derived decellularized extracellular matrix (bdECM) and tricalcium phosphate (TCP), individually and in combination, as an osteogenic promoter between bone and a 3D-printed PCL scaffold in rat calvarial defects. The diameter of the scaffolds, the diameters of the pores and the scaffold porosity were similar to the parameters of our scaffolds (6 mm for both scaffolds, 300 vs. 240 µm, 52% vs. 50%, respectively), but the implantation time was shorter (4 weeks vs. 8 weeks). In this study, histopathological cross-sections of scaffolds were also not presented, even though µCT cross-sections were analyzed and compared only with longitudinal, i.e., not “true” histopathological cross-sections, which could have been more helpful. The image of the scaffold also did not depict its fine trabecular structure, and visible fibers were distorted, swollen, and crushed adjacent tissues [36]. 

The Technovit technique used in both studies (Technovit^®^ 9100 Methyl Methacrylate) is a commercially available plastic embedding system developed for the histopathological preparation of mineralized and soft tissues [37]. Its protocol differs from that presented in this paper in relation to the embedding procedure. Instead of using paraffin with a low melting point (49 °C), polymerization was conducted using special kits at a processing temperature of +4 °C. The use of a temperature lower than the glass transition temperature of PCL (58 °C) in this process is fully consistent with our assumptions. During the development of our method, we found that the temperature close to the melting point of PCL causes expansion and swelling of the scaffold fibers, decreasing the pore size (Figure 5). Therefore, reducing the temperature below this point is justified and avoids the distortion of the scaffold [38].

Another difference was the use of xylene in the clearing stage of the protocol. We replaced xylene with d-limonene because we proved that it acted destructively on the PCL fibers, causing them to dissolve, swell, and close the pore lumen (Figure 4). 

The destructive effect of xylene on PCL has also been confirmed in other studies [39].

In our opinion, the lack of elimination of xylene in the clearing process may be the cause of the suboptimal results obtained by this method. Replacing it with d- limonene, for example, obtained from the distilled and stabilized light fractions of oils from orange peel and maize kernels, could solve this problem [40]. Eliminating xylene should not only concern the clearing process, but also the remaining stages of the histopathological protocol, such as cover slipping where xylene-soluble glue is used. In our protocol, we eliminate this problem through the use of Canadian balm glue, which is soluble in d-limonene. 

In addition, the polymers obtained in the Technovit method must be processed using the hard-cutting or division thin section technique with the use of glass and diamond knives. This may cause a problem with obtaining thin sections showing the scaffold layers and pore distributions in each layer. In our method, the thickness of the sections may be lower because embedded paraffin is more plastic than polymers; however, to avoid technical problems in obtaining very thin sections, salinized spot plates of increased adherence should be used during the transferring stage. 

Similar observations and conclusions can be reached by comparing numerous articles in which the standard histopathological technique was used. For example, Nulty et al. implanted similar PCL scaffolds (4 mm in diameter, 5 mm height) in critical-sized defects created in rat femurs. After 6 and 12 weeks, they evaluated the results using a standard histopathological technique with xylene and standard paraffin. The obtained histopathological images presented in their study showed only small representative scaffold fragments, and only its longitudinal cross-section. Scaffold fibers were also fused, melted, and swollen, and the trabecular structure of the scaffold was not visible [31]. 

The major limitation of the study is the lack of its comparison with alternative methods, including the Technovit method mentioned above. This is because this method was not known at the time of this study’s conceptualization. Another limitation is the evaluation of only one histopathological staining technique (HE). It is necessary to evaluate this protocol in more histopathological staining techniques. Thus far, our method has been used with success in the histochemical (von Kossa, Masson Trichrome,) and immunohistochemical (CD31, Factor VIII-related antigen, SATB2) staining of PCL and PCL/PLGA/β TCP scaffolds. 

In the future, we plan to compare our protocol with the Technovit technique and other alternatives and evaluate its application in a wider spectrum of staining techniques and scaffold materials. We hope that the propagation of this method will enable the determination of its limitations, and eventual modifications, depending on the requirements. 

## 5. Conclusions

The presented PCL scaffold histopathological protocol is cheap, reproducible, and enables a comprehensive evaluation of PCL scaffolds while maintaining their trabecular honeycomb-like cross-sectional structure. It also enables very thin sections showing the scaffold layers and pore distribution in each layer to be obtained. Moreover, obtaining full cross-section views of small scaffolds, as well as histochemical and immunohistochemical staining procedures, is possible using this method. Further studies are needed to assess the usefulness and limitations of this method in the histopathological preparation of tissue-engineered products. 

## Figures and Tables

**Figure 1 materials-15-01732-f001:**
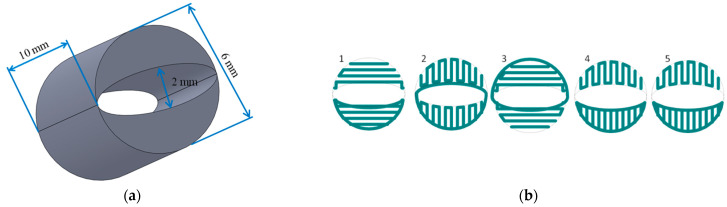
Scaffold design: (**a**) scaffold dimensions; (**b**) the fiber pattern repeated every five layers during scaffold printing.

**Figure 2 materials-15-01732-f002:**
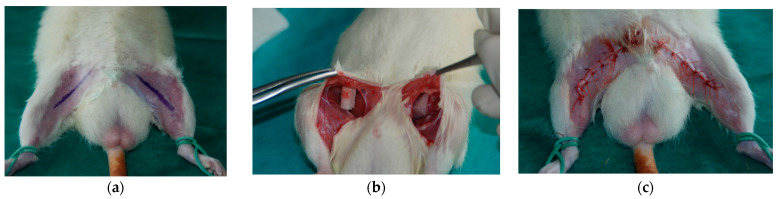
Scaffold implantation: (**a**) incision marking after the epilation of inguinal areas; (**b**) implanted scaffolds; (**c**) closed wound after implantation.

**Figure 3 materials-15-01732-f003:**
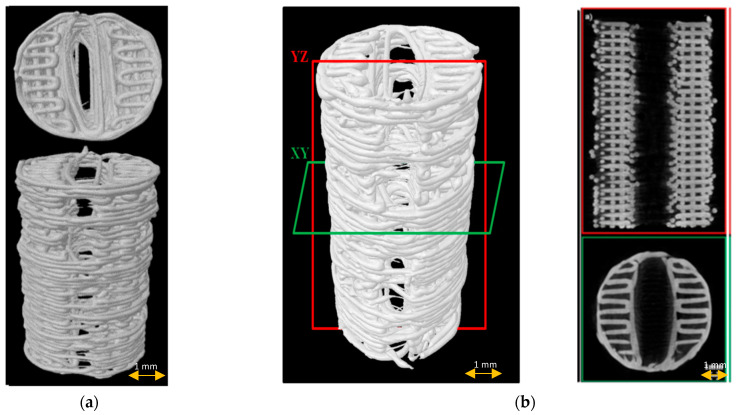
Scaffold µCT imaging: (**a**) 3D scaffold reconstruction; (**b**) µCT images of scaffold longitudinally (red frame) and cross-sectionally (green frame).

**Figure 4 materials-15-01732-f004:**
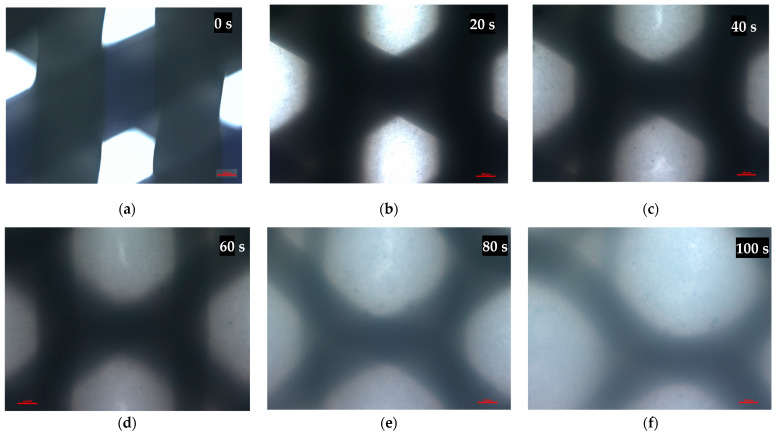
Microscopical images of the scaffold pores before (**a**) and after (**b**–**f**) xylene addition at a temperature of 23 °C. Representative images are shown at 10× magnification. Images were taken in 20-s intervals. Scale bars represent 100 μm.

**Figure 5 materials-15-01732-f005:**
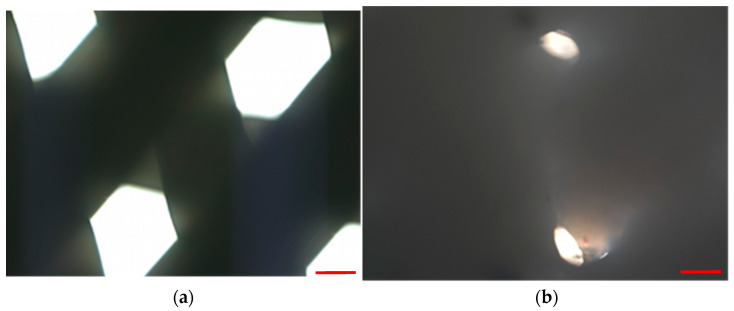
Microscopical images of scaffold pores before (**a**) and after bathing in water at a temperature of 57 °C for 10 min. Representative images are shown at 10× magnification. Scale bars represent 100 μm.

**Figure 6 materials-15-01732-f006:**
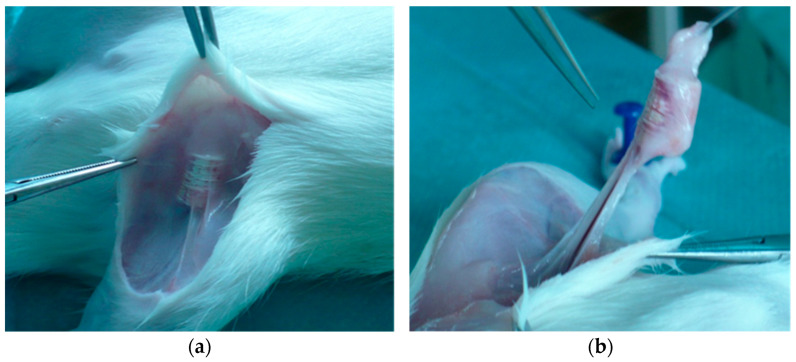
Scaffold harvesting after 2 months of follow-up (**a**,**b**).

**Figure 7 materials-15-01732-f007:**
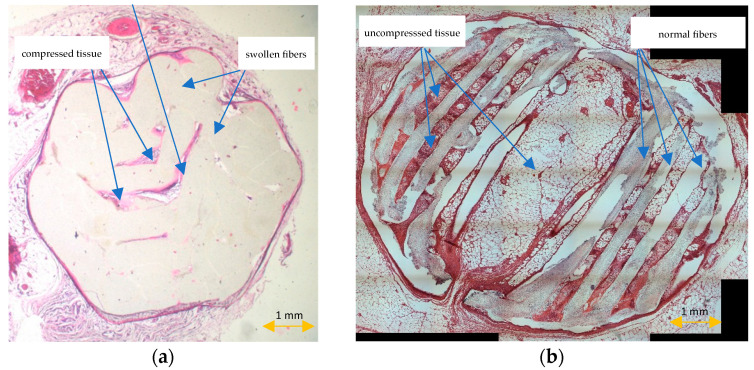
HE-stained sections prepared according to the standard (**a**) and modified histopathological protocol (**b**).

**Figure 8 materials-15-01732-f008:**
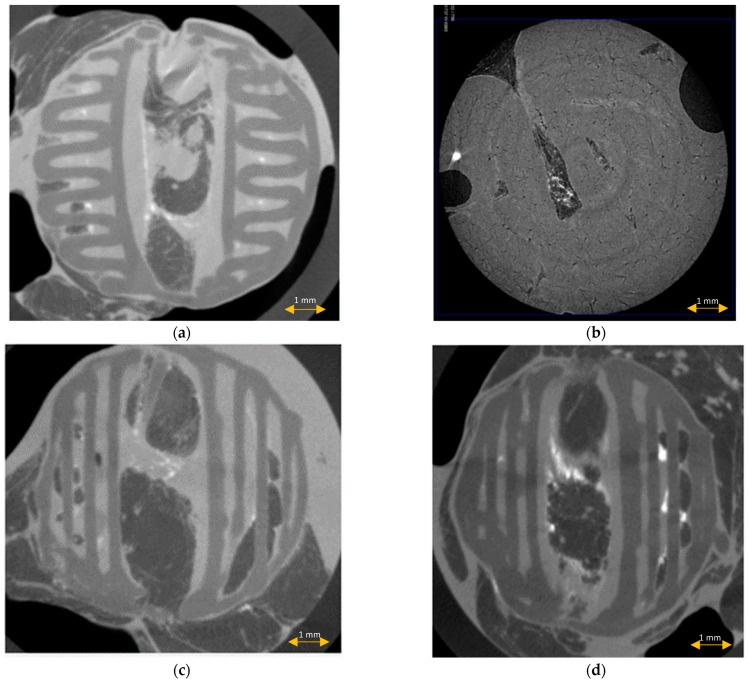
μCT imaging of the scaffolds before (**a**) and after the standard (**b**) and modified histopathological protocol ((**c**,**d**), accordingly).

**Figure 9 materials-15-01732-f009:**
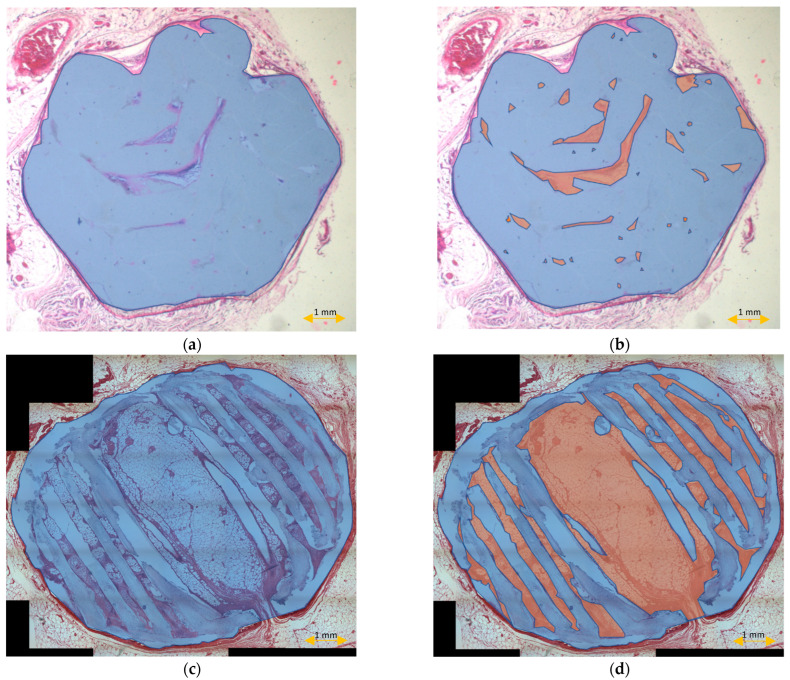
Scaffold cross-section surface (S1, blue) and soft tissue area inside the scaffold (S2, orange) on sections obtained after the standard histopathological (**a**,**b**) and modified histopathological protocols (**c**,**d**).

**Figure 10 materials-15-01732-f010:**
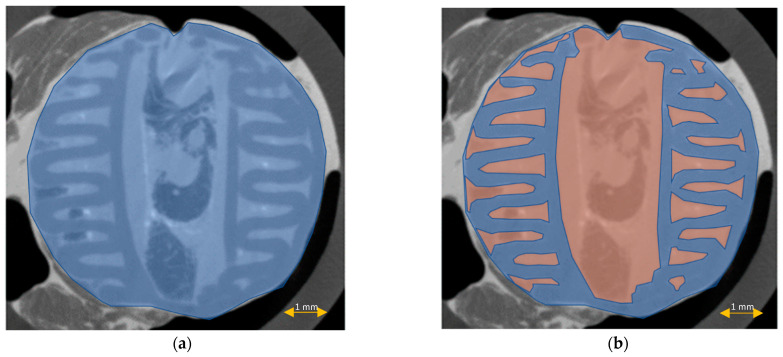
Scaffold cross-section surface (S1, blue) and soft tissue area inside the scaffold (S2, orange) on μCT images obtained before the standard histopathological (**a**,**b**) and modified histopathological protocols (**c**,**d**).

**Figure 11 materials-15-01732-f011:**
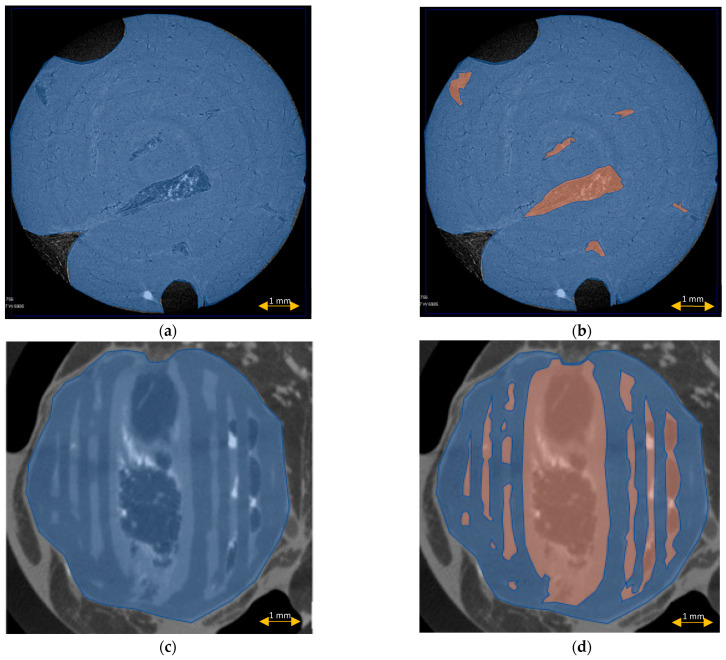
Scaffold cross-section surface (S1, blue) and soft tissue area inside the scaffold (S2, orange) on μCT images obtained after the standard histopathological (**a**,**b**) and modified histopathological protocols (**c**,**d**).

**Figure 12 materials-15-01732-f012:**
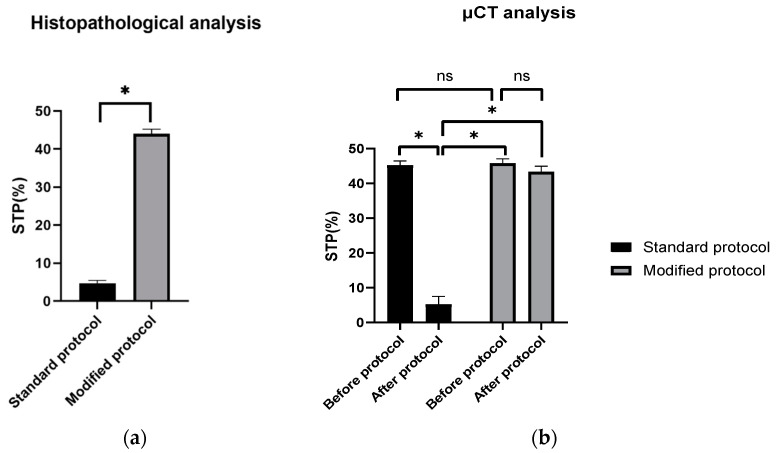
STP calculated on: (**a**) H–E sections after standard and modified histopathological protocols. Significant difference: *p* < 0.05 (*); (**b**) μCT images obtained before and after the standard and modified histopathological protocols. Significant difference: *p* < 0.05 (*), *p* > 0.05 (ns).

**Table 1 materials-15-01732-t001:** Settings of the 3D printer [28].

Settings	Value
The speed of the needle in the XY plane	80 rpm
The speed of the needle travel along the *Z* axis	400 rpm
Pneumatic pressure	5 Bar
Screw rotation speed	105 rpm
Distance between fiber layers	200 μm
Distance between parallel fibers	500 μm
Needle diameter	250 μm (G25)
Temperature	100 °C

**Table 2 materials-15-01732-t002:** Standard and new histopathological protocols.

Stage	Standard Histopathological Protocol	New Histopathological Protocol
	Ethanol 70%	Ethanol 70%
Dehydration	Ethanol 90%	Ethanol 90%
	Ethanol 99.8%	Ethanol 99.8%
	Ethanol 99.8%	Ethanol 99.8%
Clearing	Xylene	HistoClear^®^
	Xylene	HistoClear^®^
	Paraffin in 57 °C	Paraffin in 49 °C
Embedding	Paraffin in 57 °C	Paraffin in 49 °C
	Embedding in paraffin blocks 58 °C	Embedding in paraffin blocks in 50 °C
Sectioning	4 µm thick sections	4 µm thick sections
Section transferring	Standard microscope slides	Highly adhesive microscope slides (Superfrost Ultra Plus^®^)
Deparaffinization	Xylene	HistoClear^®^
Rehydration	Reversed clearing (Xylene) and dehydration (graded Ethanol) Water rinsing	Reversed clearing (HistoClear^®^) dehydration (graded Ethanol) Water rinsing
HE staining	Hematoxylin (2 min) Eosin (30 s) Water rinsing	Hematoxylin (2 min) Eosin (30 s) Water rinsing
Dehydration	As above	As above
Cover slipping	Xylene-based glue	Canadian balm soluble in Histoclear^®^

**Table 3 materials-15-01732-t003:** μCT imaging settings.

Settings	Value
Source voltage	40 kV
Source current	250 μA
Rotation step	0.4° (up to 180°)
Exposure time	100 ms
Projection count for radiograph averaging	5 projections
Random movement	50 μm
Image-pixel size	9.93 μm

## Data Availability

Not applicable.

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
