# Peer review of "Modified Histopathological Protocol for Poly-ɛ-Caprolactone Scaffolds Preserving Their Trabecular, Honeycomb-like Structure"

_materials, 2022, doi:10.3390/ma15051732_

Round 1

Reviewer 1 Report

In this work, the authors developed a standard histopathological protocol for PCL scaffold preparation and evaluate it on porous cylindrical PCL scaffolds in a rat model. They evaluate the standard and new histopathological protocols, and after that perform the Scaffold μCT imaging. The topic is interesting. The title is in accordance with the subject dealt with in the manuscript. The abstract has an adequate length and clearly expresses the objectives and main results obtained in the study. However, there are some drawbacks to the article but the study should be published in a journal after incorporating the following modifications:

1- the manuscript need high grammatical and spelling editing

2- equation numbering is lost in the manuscript.

3- I suggest extending and using more keywords.

4- In the scaffold fabrication section, the table 1 is based on which references?

5- the author should extend the conclusion part.

Reviewer 2 Report

The manuscript is dealing with the designed poly-É›-caprolactone (PCL) nanomaterials, porous cylindrical PCL scaffolds, to elaborate  more sustainable  histopathological protocol

The present manuscript reflects a comprehensive study starting from up-to-date 3D printing methods of preparation of designed scaffolds (characterized by means of X-ray microtomography) and re-designing of the animal study followed by the new histopathological protocol. In essence, the new protocol excludes xylene, a non-desirable solvent by replacing with biodegradable commercially available HistoClear. The presented PCL scaffold histopathological protocol is confirmed by the authors as cheap, reproducible, and suitable for comprehensive evaluation of PCL scaffolds with maintaining the honeycomb - like structure on cross-sections. The language of the manuscript is correct and concise, figures and tables provide a sufficient complementary data to prove the author’s message.

In general, the paper looks very thoroughly and can be accepted in its current version, except one concern that requires a minor revision. 

 I would recommend paying attention to the following aspect prior to have its accepted. Telling about biodegradability, authors cited a paper of 2007, a recognized but a bit old work. Since then, plenty of work have been published on the biodegradability and toxicity of PCL and misc composites with it. PCL is considered to be not rapidly but still biodegradable material, and number 1-3 years is too general and may not be applied to this material. For the materials lie PCL, it may depend on the environmental conditions or a pretreatment applied to the used samples. I would recommend adding more recent work to prove the low toxicity of the material. Another issue could be size and shape of the nanomaterial, which may make difference in terms of toxicity. How authors can prove that this size/shape are not toxic for normal cells? A rf related to the study about toxicity of the nanomaterials with these or similar properties is desirable

Reviewer 3 Report

It is a very interesting work that presents a new protocol to maintain the stability of implants of a pcl base during the process of histological paraffin preparation. However, some changes are being suggested for better presentation of the article.

In the introduction, “Standard histological protocols [20] can destroy carefully created 3-d, honeycomb like architecture of PCL scaffolds and lead to scaffold fibers swelling resulting in displacement or compression of tissues inside scaffold”, it is necessary to add a reference to the statement. Perhaps use this information in the discussion part of the study detailing the physicochemical interactions that occur in pcl scaffolds with the use of different protocols (or specific reagents for each protocol that can change the conformation of the biomaterial).

In figure 5 presented in the results, indicate the structures for better identification of them. It would be interesting to add a photo of each protocol applied to better demonstrate the structures described in the text.

Is there any work that has used histological processing with the inclusion of pcl scaffolds in paraffin? Add some article in the discussion for better comparison between studies.

In the discussion it was mentioned that STP had a difference of 1.5% between the uCT and histological analysis. If possible, add this information to results.

Figure 10 and 11 correspond to an initial test for the development of the modified histology protocol. As this is part of the results, describe in the results part and add the methodology used in the materials and methods section.